# Curcumin Alleviates Aflatoxin B_1_-Induced Liver Pyroptosis and Fibrosis by Regulating the JAK2/NLRP3 Signaling Pathway in Ducks

**DOI:** 10.3390/foods12051006

**Published:** 2023-02-27

**Authors:** Yilong Cui, Qi Wang, Xuliang Zhang, Xu Yang, Yun Shi, Yanfei Li, Miao Song

**Affiliations:** 1College of Animal Science and Technology, Inner Mongolia Minzu University, Tongliao 028000, China; 2Key Laboratory of the Provincial Education, Department of Heilongjiang for Common Animal Disease Prevention and Treatment, College of Veterinary Medicine, Northeast Agricultural University, Harbin 150030, China; 3College of Veterinary Medicine, Henan Agricultural University, Zhengzhou 450002, China; 4Tongliao City Animal Quarantine Technical Service Center, Tongliao 028000, China

**Keywords:** dietary curcumin, AFB_1_ exposure, hepatocyte pyroptosis, liver fibrosis, Jinding duck

## Abstract

Aflatoxin B_1_ (AFB_1_) is a serious pollutant in feed and food which causes liver inflammation, fibrosis, and even cirrhosis. The Janus kinase 2 (JAK2)/signal transducers and activators of the transcription 3 (STAT3) signaling pathway is widely involved in inflammatory response and promotes the activation of nod-like receptor protein 3 (NLRP3) inflammasome, thus leading to pyroptosis and fibrosis. Curcumin is a natural compound with anti-inflammatory and anti-cancer properties. However, whether AFB_1_ exposure leads to the activation of the JAK2/NLRP3 signaling pathway in the liver and whether curcumin can regulate this pathway to influence pyroptosis and fibrosis in the liver remains unclear. In order to clarify these problems, we first treated ducklings with 0, 30, or 60 µg/kg AFB_1_ for 21 days. We found that AFB_1_ exposure caused growth inhibition, liver structural and functional damage, and activated JAK2/NLRP3-mediated liver pyroptosis and fibrosis in ducks. Secondly, ducklings were divided into a control group, 60 µg/kg AFB_1_ group, and 60 µg/kg AFB_1_ + 500 mg/kg curcumin group. We found that curcumin significantly inhibited the activation of the JAK2/STAT3 pathway and NLRP3 inflammasome, as well as the occurrence of pyroptosis and fibrosis in AFB_1_-exposed duck livers. These results suggested that curcumin alleviated AFB_1_-induced liver pyroptosis and fibrosis by regulating the JAK2/NLRP3 signaling pathway in ducks. Curcumin is a potential agent for preventing and treating liver toxicity of AFB_1_.

## 1. Introduction

Aflatoxin B_1_ (AFB_1_) is one of the most harmful mycotoxins in feed and food [1]. In 1996, the World Health Organization set the upper limit of AFB_1_ in cereals entering the market at 20 μg/kg [2]. However, many parts of the world have remained heavily contaminated with AFB_1_ for many years, posing a threat to animal and human health. In 2014, Iram et al. found that the average content of AFB_1_ in 487 poultry feeds and raw materials in Pakistan was 37.62 μg/kg and 23.75 μg/kg, respectively [3]. In 2019, Akinmusire et al. found that 101 poultry feeds and raw materials in Nigeria contained an average of 74 μg/kg AFB_1_ [4]. In 2019, Mahuku et al. found that the average AFB_1_ content in maize samples from eastern and southwestern Kenya was 67.8 μg/kg and 22.3 μg/kg, respectively [5]. Long-term ingestion of food or feed contaminated with AFB_1_ can lead to chronic poisoning in humans and animals, with symptoms such as dizziness, anorexia, convulsions, and loss of memory function [6]. Chronic AFB_1_ poisoning can cause damage to the liver, kidney, spleen, testis, and other parenchymal organs, resulting in growth disorder, immunosuppression, hematopoietic damage, and reduced reproductive performance [7,8,9,10]. Therefore, the question of how to alleviate the harm caused by AFB_1_ has become a hot issue in today’s research.

The liver is the main organ of metabolism and detoxification and is also the main target organ for AFB_1_ to exert toxicity [11]. AFB_1_ accumulation in the liver can cause inflammation and fibrosis, and the progression of fibrosis may lead to cirrhosis [12]. Recently, some studies have reported that the nod-like receptor protein 3 (NLRP3) inflammasome is associated with liver fibrosis [13,14]. Once the toxin enters the liver, NLRP3 molecules collect apoptosis-associated speck-like protein (ASC) and pro-caspase-1 molecules to assemble into the NLRP3 inflammasome, which catalyzes pro-caspase-1 to become mature cysteinyl aspartate-specific proteinase 1 (Caspase-1) [15]. Subsequently, mature Caspase-1 catalyzes the hydrolysis of interleukin-1β (IL-1β) precursor protein and interleukin-18 (IL-18) precursor protein into mature IL-1β and IL-18, which are released into the extracellular matrix, triggering an inflammatory response and promoting fibrosis progression [16]. In addition, Caspase-1 can cleat the Gasdermin D (GSDMD) protein into GSDMD-N, which can cause pyroptosis through oligomerization on the cell membrane and further promote fibrosis [17]. The Janus kinase 2 (JAK2)/signal transducers and activators of the transcription 3 (STAT3) pathway is a newly discovered pathway widely involved in inflammation, pyroptosis, and fibrosis, which has been proved to activate the NLRP3 inflammasome [18]. However, it is unclear whether AFB_1_-induced liver pyroptosis and fibrosis is related to the JAK2/NLRP3 signaling pathway.

AFB_1_ poisoning is more common in chickens, ducks, pigs, and other economic animals, among which ducks are the most susceptible, with young and male animals being more susceptible than adult and female animals [19]. In 1977–1978, AFB_1_ poisoning caused sudden concentrated deaths of waterfowl in two areas of Texas [20]. In addition, after livestock and poultry eat feed contaminated with AFB_1_, the toxin can still remain in meat, eggs, milk, and other animal-derived food, and bring great safety risks to human health through the food chain [21]. Some scholars estimate that AFB_1_ pollution costs the United States between USD 52.1 million and USD 1.68 billion annually [22].

Curcumin, a diketone compound, is a plant polyphenol extracted from the rhizome of *Curcuma longa* [23]. This natural compound, which is widely found in Sichuan and Guizhou provinces of China, has been shown to have antiviral, antifungal, and anti-cancer pharmacological effects [24]. Although curcumin is rapidly metabolized in the body, and ways to improve its bioavailability are still being sought, its effective effects on neurological diseases, cardiovascular disease, and diabetes have been widely reported [25]. In animal studies, curcumin can improve the antioxidant and anti-inflammatory ability of ducks, reduce the liver and intestinal damage caused by AFB_1_ and ochratoxin A, and thus improve the performance of ducks, and it has been confirmed that it is safe and harmless for ducks to feed on curcumin alone [26,27]. However, whether curcumin can alleviate liver pyroptosis and fibrosis caused by AFB_1_ has not been reported.

Therefore, the purpose of this study was to explore whether curcumin could alleviate AFB_1_-induced liver pyroptosis and fibrosis by regulating the JAK2/NLRP3 signaling pathway in ducks, so as to provide an experimental basis for alleviating the liver toxicity of AFB_1_ in clinic.

## 2. Materials and Methods

### 2.1. Schematic Overview of Experimental Program

Figure 1 describes the design idea of this study.

### 2.2. Animals and Treatment

In the first part of the experiment, one-day-old healthy male Jinding ducks (Anas platyrhyncha) were randomly divided into three groups (*n* = 6). The dose of AFB_1_ was determined according to previous studies [28,29] and the median lethal dose in ducks (LD_50_ = 300 µg/kg body weight) [30]. AFB_1_ (≥99.8%, Qingdao Pribolab Pte. Ltd., Qingdao, China) was dissolved in corn oil. The low-dose group (LG) and high-dose group (HG) were given 30 µg/kg (1/10 LD_50_) and 60 µg/kg (1/5 LD_50_) body weight of AFB_1_ by intragastric administration daily, respectively. The control group (CG) was given the same volume of solvent. The experiment lasted three weeks.

We found that 60 µg/kg AFB_1_ caused more significant fibrotic lesions in the duck liver, so this dose was selected for the second part of the experiment. One-day-old healthy male ducks were divided into three groups (*n* = 6): control group (CG), AFB_1_ exposure group (AG, 60 µg/kg AFB_1_), and AFB_1_ + curcumin group (ACG, 60 µg/kg AFB_1_ and 500 mg/kg curcumin). Curcumin (≥99.8%, Nanjing NutriHerb BioTech Co., Ltd., Nanjing, China) was added to the base diet at doses determined by previous studies [27,31]. The experiment lasted three weeks.

All experimental routines and protocols have been approved by the Animal Ethics Committee of the Northeast Agricultural University (NEAUEC20230332). The ducks were kept in the Biomedical Research Center of Northeast Agricultural University and had free access to water and pellet feed formulated according to the National Research Council. The temperature in the room was 32 °C for the first week, 30 °C for the second week, and 28 °C for the third week, guaranteeing 18 h of incandescent light a day. No animals died during the experiment.

### 2.3. Sample Collection

After stopping feeding overnight (12 h) on the 21st day, the ducks were weighed; then, the blood was collected through the wing vein and serum was collected after centrifugation at 1500× *g* for 10 min. The ducks were anesthetized by intravenous injection of sodium pentobarbital (50 mg/kg body weight). Livers were quickly collected, one part was fixed with 4% formaldehyde for histopathological examination, and the rest was stored at −80 °C for other studies.

### 2.4. Measurement of Serum ALT and AST Activities

Serum alanine aminotransferase (ALT) and aspartate aminotransferase (AST) activities were measured using an automatic hematology analyzer.

### 2.5. Histopathological Observation

The liver was fixed in 4% paraformaldehyde for 24 h, then embedded in paraffin and sliced. Hematoxylin eosin (HE) staining and sirius red staining were performed, respectively, according to previous studies [32]. The sections were scanned and then observed with CaseViewer software (3Dhistech, Budapest, Hungary) to evaluate the degree of histopathological damage and fibrosis.

### 2.6. TUNEL Analysis

According to the instruction of the biochemical kit (Beyotime, Shanghai, China), the above liver sections were stained using terminal deoxynucleoside transferase (TdT) dUTP nick end labeling (TUNEL). The sections were scanned and then observed with CaseViewer software (3Dhistech, Budapest, Hungary) to evaluate the liver pyroptosis level.

### 2.7. Detection of IL-1β and IL-18 Levels in the Liver and Serum

The 50 mg liver sample was added to 1 mL phosphate-buffered saline, fully ground with homogenizer, and then centrifuged at 12,000× *g* for 10 min to collect the supernatant. The contents of IL-1β and IL-18 in the supernatant and serum were determined using the ELISA kits (Nanjing Jiancheng, Nanjing, China).

### 2.8. RT-qPCR Analysis

The total RNA of duck liver was extracted according to the instruction of Trizol reagent (Invitrogen, Carlsbad, CA, USA), and then the cDNA was synthesized using the reverse transcription kit (Roche, Basel, Switzerland). The mRNA expression was detected with an ABI PCR system (Thermo Fisher, Waltham, MA, USA). β-actin was selected as the internal reference gene, and the relative mRNA expression was calculated using the 2^-ΔΔCt^ method. All the information on primers is shown in Appendix A.

### 2.9. Western Blot Analysis

The 100 mg liver sample was added to 1 mL RIPA lysate (Beyotime, Shanghai, China) with 10 μL PMSF (Beyotime, Shanghai, China), fully ground with homogenizer, then centrifuged at 12,000× *g* for 10 min at 4 °C to collect the supernatant. A BCA kit (Solarbio, Beijing, China) was used to detect the protein concentration, and 30 μg protein samples were selected for 5−12% SDS-PAGE gel electrophoresis. The gel was then transferred to the PVDF membrane, incubated overnight with the required primary antibody at 4 °C, and then incubated with the corresponding secondary antibody for 1 h at 37 °C. Images were collected in a gel imaging system (General Electric, Fairfield, CT, USA) using an ECL luminescence solution and analyzed using Image J software (National Institutes of Health, Bethesda, MD, USA). All the information on antibodies is shown in Appendix A.

### 2.10. Statistical Analysis

Data were analyzed using SPSS 25.0 software (SPSS Incorporated, Chicago, IL, USA) and expressed as mean ± standard deviation (mean ± SD). * and ** represent *p* < 0.05 and *p* < 0.01 vs. the CG, respectively; # and ## represent *p* < 0.05 and *p* < 0.01 vs. the AG, respectively.

## 3. Results

### 3.1. AFB_1_ Exposure Caused Liver Damage in Ducks

Compared with the CG, the body weight decreased significantly by 11% and 24.5%, the serum ALT activities increased significantly by 41% and 87%, and the serum AST activities increased significantly by 39% and 76% in the LG and HG, respectively (Figure 2A–C). HE staining showed that the CG liver had normal microscopic structure, while the LG and HG livers showed hepatic cord disorders, inflammatory cell infiltration, and even hepatic cell disintegration (Figure 2D).

### 3.2. AFB_1_ Exposure Activated JAK2/NLRP3-Mediated Pyroptosis in Duck Livers

The mRNA expressions of JAK2 and STAT3 in the LG were 1.74 and 2 times that in the CG, respectively, and the mRNA expressions of JAK2 and STAT3 in the HG were 2.74 and 3.04 times that in the CG, respectively (Figure 3A). The protein expressions of pJAK2/JAK2 and pSTAT3/STAT3 in the LG were 1.76 and 2.41 times that in the CG, respectively, and the protein expressions of pJAK2/JAK2 and pSTAT3/STAT3 in the HG were 3.06 and 4.59 times that in the CG, respectively (Figure 3B). Compared with the CG, TUNEL staining showed that the positive rate of the LG and HG increased from 1.43% to 10.6% and 20%, respectively (Figure 3C,D). The mRNA expressions of NLRP3, ASC, and Caspase-1 in the LG were 2.87, 2.56, and 1.95 times that in the CG, respectively, and the mRNA expressions of NLRP3, ASC, and Caspase-1 in the HG were 4.37, 4.23, and 4.43 times that in the CG, respectively (Figure 3E). The protein expressions of NLRP3, ASC, Caspase-1, GSDMD, and GSDMD-N in the LG were 1.36, 2.19, 1.59, 1.89, and 2.58 times that in the CG, respectively, and the protein expressions of NLRP3, ASC, Caspase-1, GSDMD, and GSDMD-N in the HG were 2.56, 3.86, 2.5, 4.15, and 8.32 times that in the CG, respectively (Figure 3F).

### 3.3. AFB_1_ Exposure Caused Liver Fibrosis in Ducks

Compared with the CG, the liver IL-1β levels increased significantly by 22% and 62%, the liver IL-18 levels increased significantly by 14% and 38%, the serum IL-1β levels increased significantly by 22% and 61%, and the serum IL-18 levels increased significantly by 15% and 34% in the LG and HG, respectively (Figure 4A,B). The mRNA expressions of IL-1β and IL-18 in the LG were 2.82 and 1.85 times that in the CG, and the mRNA expressions of IL-1β and IL-18 in the HG were 4.4 and 2.67 times that in the CG, respectively (Figure 4C). Sirius red staining showed that the liver fibrosis of ducks became more and more obvious with the gradual increase in AFB_1_ exposure dose (Figure 4D). The mRNA expressions of α-SMA, Col-I;, and TGF-β in the LG were 2.39, 2.22, and 1.79 times that in the CG, and the mRNA expressions of α-SMA, Col-I;, and TGF-β in the HG were 4.08, 4.33, and 3.33 times that in the CG, respectively (Figure 4E). The protein expressions of α-SMA, Col-I;, and TGF-β in the LG were 2.5, 2.12, and 1.53 times that in the CG, and the protein expressions of α-SMA, Col-I;, and TGF-β in the HG were 4.15, 3.6, and 2.33 times that in the CG, respectively (Figure 4F).

### 3.4. Curcumin Alleviated Liver Damage in Ducks Caused by AFB_1_ Exposure

Compared with the AG, the body weight increased significantly by 19%, the serum ALT activity decreased significantly by 24%, and the serum AST activity decreased significantly by 21% in the ACG, respectively (Figure 5A–C). Compared with the AG, HE staining showed significant remission of liver lesions in the ACG (Figure 5D).

### 3.5. Curcumin Alleviated JAK2/NLRP3-Mediated Pyroptosis in the Liver of Ducks Exposed to AFB_1_

The mRNA expressions of JAK2 and STAT3 in the ACG were 0.61 and 0.58 times that in the AG, respectively (Figure 6A). The protein expressions of pJAK2/JAK2 and pSTAT3/STAT3 in the ACG were 0.74 and 0.71 times that in the AG, respectively (Figure 6B). Compared with the AG, TUNEL staining showed that the positive rate of ACG decreased from 18% to 9.4% (Figure 6C,D). The mRNA expressions of NLRP3, ASC, and Caspase-1 in the ACG were 0.71, 0.53, and 0.6 times that in the AG, respectively (Figure 6E). The protein expressions of NLRP3, ASC, Caspase-1, GSDMD, and GSDMD-N in the ACG were 0.79, 0.68, 0.63, 0.65, and 0.65 times that in the AG, respectively (Figure 6F).

### 3.6. Curcumin Alleviated Liver Fibrosis in Ducks Caused by AFB_1_ Exposure

Compared with the AG, the liver IL-1β and IL-18 levels decreased significantly by 24% and 17%, respectively, and the serum IL-1β and IL-18 levels decreased significantly by 27% and 13% in the ACG, respectively (Figure 7A,B). The mRNA expressions of IL-1β and IL-18 in the ACG were 0.63 and 0.62 times that in the AG, respectively (Figure 7C). Compared with the AG, sirius red staining showed significant remission of liver fibrosis in the ACG (Figure 7D). The mRNA expressions of α-SMA, Col-I;, and TGF-β in the ACG were 0.76, 0.54, and 0.67 times that in the AG, respectively (Figure 7E). The protein expressions of α-SMA, Col-I;, and TGF-β in the ACG were 0.85, 0.73, and 0.67 times that in the AG, respectively (Figure 7F).

## 4. Discussion

In this study, we first found that AFB_1_ exposure resulted in slow growth, liver structural and functional impairment, and caused the JAK2/NLRP3-mediated liver pyroptosis and fibrosis in ducks. Secondly, we found that curcumin alleviated AFB_1_-induced liver pyroptosis and fibrosis by regulating the JAK2/NLRP3 signaling pathway in ducks. These results provide a new understanding for exploring the mechanism of hepatotoxicity of AFB_1_ and provide an experimental basis for curcumin to alleviate the toxicity of AFB_1_.

AFB_1_ has the highest accumulation in liver after entering the body [33]. Meissonnier et al. found that AFB_1_ exposure resulted in decreased weight gain and dysfunction of liver structure and function in pigs [34]. Gao et al. found that AFB_1_ exposure resulted in liver damage in chickens, including hepatocyte destruction, swelling, and inflammatory cell infiltration, as well as increased ALT and AST activities [35]. Consistent with these results, we also demonstrated that AFB_1_ exposure caused slow growth and liver histopathological damage and dysfunction in ducks (Figure 2). These studies indicate that AFB_1_ is harmful to a variety of species, but the specific mechanism of liver toxicity of AFB_1_ still needs to be further explored.

Liver fibrosis is a repair reaction after liver injury which can lead to cirrhosis or even liver cancer if it continues to develop excessively [36]. During liver inflammation, hepatic stellate cells are activated and converted into myofibroblasts, which produced large amounts of extracellular matrix leading to fibrosis [37]. The increased expression of α-SMA and Col-I is the main characteristic of hepatic stellate cell activation, and TGF-β can promote myofibroblast proliferation [38]. In this study, we first found that AFB_1_ exposure caused significant fibrosis in duck livers through sirius red staining (Figure 4D). RT-qPCR and Western blot detection revealed that the expression levels of α-SMA, Col-I, and TGF-β were significantly increased, which further confirmed the occurrence of fibrosis (Figure 4E,F). AFB_1_ exposure has been shown to cause liver fibrosis in Oncorhynchus mykiss and rats in previous studies [39,40]. However, there are few studies on the specific mechanism of liver fibrosis induced by AFB_1_.

Inflammation is undoubtedly a key factor leading to fibrosis. As an intracellular protein complex, the NLRP3 inflammasome is a major inflammatory factor [41]. The activation of the NLRP3 inflammasome, on the one hand, produces IL-1β and IL-18, leading to liver fibrosis. On the other hand, it activates Caspase-1-mediated pyroptosis to promote fibrosis development [42]. In this study, we found that AFB_1_ exposure led to increased expressions of NLRP3, ASC, Caspase-1, GSDMD, and GSDMD-N in duck livers, and combined with TUNEL staining analysis, this confirmed the occurrence of pyroptosis (Figure 3C–F). In addition, the JAK2/STAT3 signaling pathway has been reported to not only activate the NLRP3 inflammasome and lead to pyroptosis, but also to promote the proliferation of hepatic stellate cells and promote fibrosis [43]. Therefore, we continued to detect the JAK2/STAT3 signaling pathway, and the results showed that this pathway was activated in the liver of AFB_1_-exposed ducks (Figure 3A,B). In conclusion, AFB_1_ exposure leads to JAK2/NLRP3-mediated pyroptosis and fibrosis in duck livers and may have a critical effect on liver damage.

The activation of the NLRP3 inflammasome directly promotes the progression of liver pyroptosis and fibrosis [44,45]. Therefore, reducing the activation of the NLRP3 inflammasome is a reliable method to reverse the development of liver pyroptosis and fibrosis and effectively alleviates liver injury. Curcumin, a naturally occurring polyphenol in turmeric, has been shown to be effective in inhibiting inflammation and pyroptosis. Yu et al. found that curcumin alleviated doxorubicin-induced cardiac injury by inhibiting NLRP3 inflammasome activation and myocardial pyroptosis [46]. In a recent study, Gan et al. found that curcumin mitigated arsenic trioxidation-induced hypothalamic damage in ducks by inhibiting neuronal pyroptosis mediated by the NF-κB/NLRP3 signaling pathway [47]. In addition, curcumin has been shown to alleviate inflammation by inhibiting the JAK/STAT signaling pathway [48,49]. Therefore, we hypothesized that curcumin could alleviate AFB_1_-induced pyroptosis and fibrosis by regulating the JAK2/NLRP3 signaling pathway in duck livers.

AFB_1_-exposed ducklings were fed a diet supplemented with curcumin for 21 days. These ducks showed improved growth rates compared to ducks exposed only to AFB_1_ (Figure 5A). HE staining showed that the damage of the liver microstructure was significantly alleviated, and the activities of ALT and AST were also decreased (Figure 5B–D). Similarly, Jin et al. demonstrated that dietary curcumin could improve the growth performance, brisket quality, and antioxidant capacity of multiple organs, as well as the damage of liver structure and function in ducks [28,50]. Moreover, we found that curcumin alleviated AFB_1_-induced duck liver fibrosis by sirius red staining and detection of α-SMA, Col-I, and TGF-β expression (Figure 7D–F). The alleviative effect of curcumin on duck liver fibrosis induced by AFB_1_ may be the key factor for curcumin to protect the liver, and we further explored the molecular mechanism.

We first detected factors related to the JAK2/STAT3 pathway and found that curcumin inhibited the activation of this pathway in the liver of AFB_1_-exposed ducks (Figure 6A,B). We then found that curcumin inhibited the increased expressions of NLRP3, ASC, Caspase-1, GSDMD, and GSDMD-N in the liver of AFB_1_-exposed ducks (Figure 6E,F). Combined with TUNEL staining analysis (Figure 6C,D), these results demonstrated that curcumin inhibited AFB_1_-induced JAK2/NLRP3-mediated pyroptosis in duck livers. It should be noted that there was no significant difference between AG in the curcumin treatment experiment and HG in the toxicity experiment in this study. These reflect the stability of the test method and the effectiveness of curcumin therapy in this study. On the whole, JAK2/NLRP3-mediated pyroptosis and fibrosis may be a key target for the prevention and treatment of liver damage induced by AFB_1_ exposure, and curcumin can regulate this pathway and alleviate the liver toxicity of AFB_1_.

## 5. Conclusions

Curcumin can alleviate AFB_1_-induced liver pyroptosis and fibrosis by regulating the JAK2/NLRP3 signaling pathway in ducks. Therefore, curcumin as a natural plant extract shows potential in preventing AFB_1_-induced liver fibrosis.

## Figures and Tables

**Figure 1 foods-12-01006-f001:**
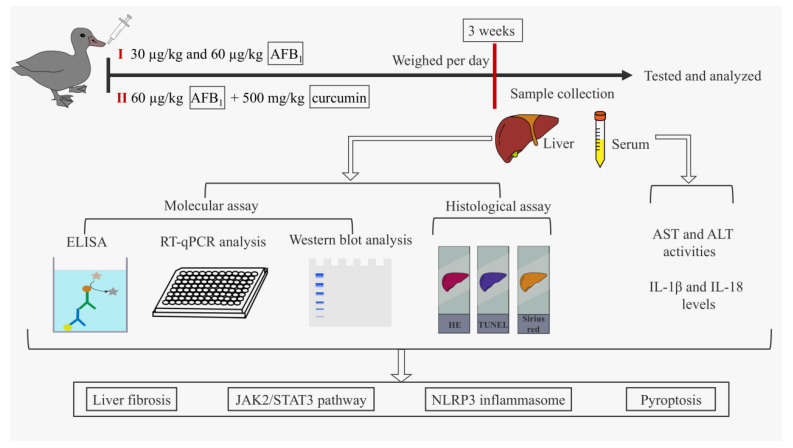
Schematic overview of experimental program. Firstly, ducks were exposed to a gradient dose of AFB_1_ for 21 days. The effects of AFB_1_ on fibrosis, JAK2/NLRP3 signaling pathway, and pyroptosis of duck livers were analyzed. Then, the dose of AFB_1_, which can cause significant liver fibrosis in ducks, was selected and treated with curcumin for 21 days. The effects of curcumin on fibrosis, JAK2/NLRP3 signaling pathway, and pyroptosis of duck livers caused by AFB_1_ were analyzed.

**Figure 2 foods-12-01006-f002:**
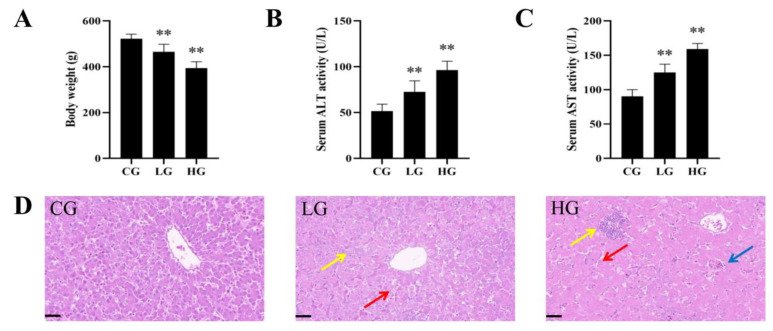
AFB_1_ exposure caused liver damage in ducks. (**A**) Body weight (*n* = 6). (**B**) Serum ALT activity (*n* = 6). (**C**) Serum AST activity (*n* = 6). (**D**) Representative images of liver HE staining (red arrow: disordered hepatic cords; yellow arrow: infiltrated inflammatory cells; blue arrow: hepatocyte disintegration). Scale bar: 50 μm. ** *p* < 0.01 vs. the CG.

**Figure 3 foods-12-01006-f003:**
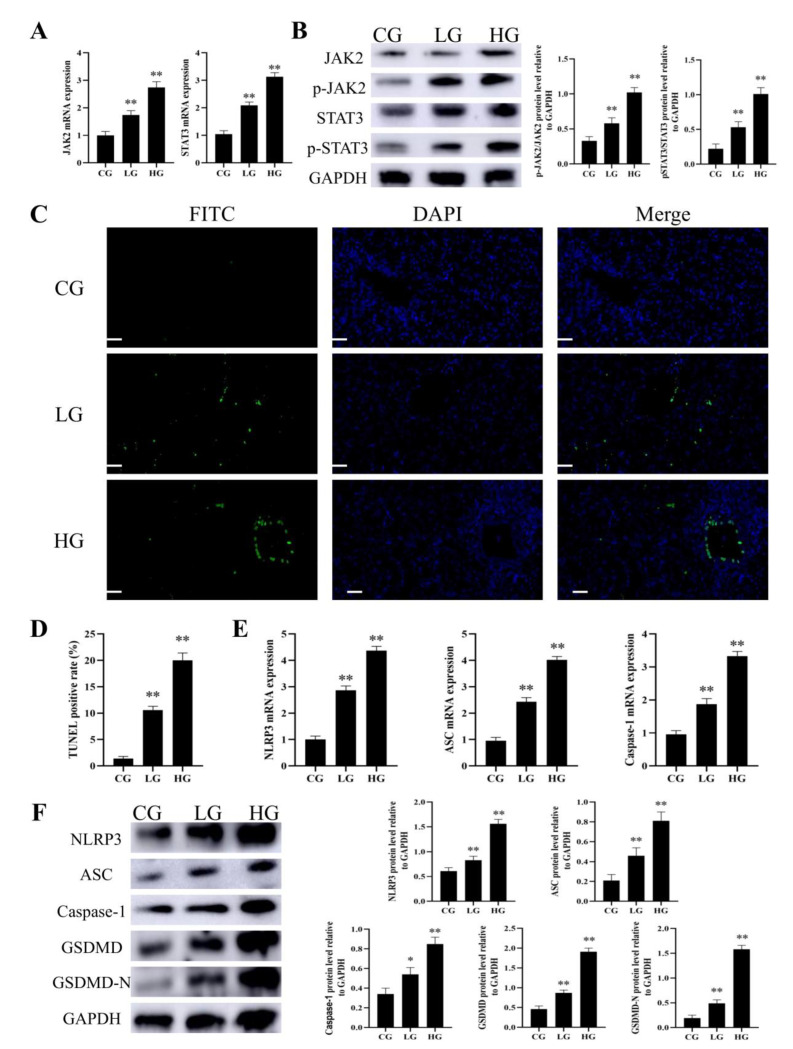
AFB_1_ exposure activated JAK2/NLRP3-mediated pyroptosis in duck livers. (**A**) mRNA expressions of JAK2 and STAT3 (*n* = 6). (**B**) Western blot of the JAK2, p-JAK2, STAT3, and p-STAT3 protein (*n* = 3). (**C**) Representative images of liver TUNEL staining. Scale bar: 50 μm. (**D**) TUNEL analysis of livers (*n* = 3). (**E**) mRNA expressions of IL-1β and IL-18 (*n* = 6). (**F**) Western blot analysis of the NLRP3, ASC, Caspase-1, GSDMD, and GSDMD-N protein levels (*n* = 3). * *p* < 0.05 and ** *p* < 0.01 vs. the CG.

**Figure 4 foods-12-01006-f004:**
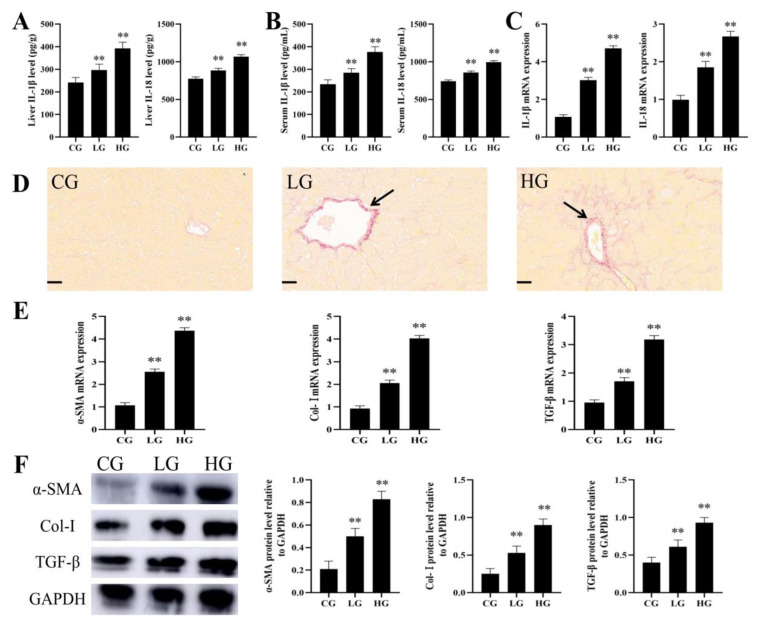
AFB_1_ exposure caused liver fibrosis in ducks. (**A**) Liver IL-1β and IL-18 levels (*n* = 6). (**B**) Serum IL-1β and IL-18 levels (*n* = 6). (**C**) mRNA expressions of IL-1β and IL-18 (*n* = 6). (**D**) Representative images of sirius red staining of duck livers (black arrow: fibrotic lesions). (**E**) mRNA expressions of α-SMA, Col-I;, and TGF-β (*n* = 6). (**F**) Western blot analysis of the α-SMA, Col-I;, and TGF-β protein levels (*n* = 3). ** *p* < 0.01 vs. the CG.

**Figure 5 foods-12-01006-f005:**
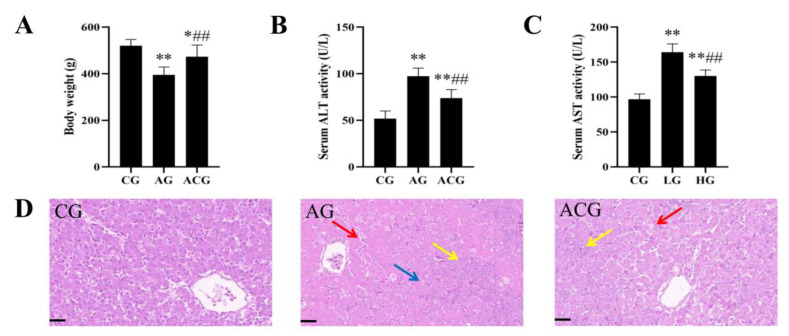
Curcumin alleviated liver damage caused by AFB_1_ in ducks. (**A**) Body weight (*n* = 6). (**B**) Serum ALT activity (*n* = 6). (**C**) Serum AST activity (*n* = 6). (**D**) Representative images of liver HE staining (red arrow: disordered hepatic cords; yellow arrow: infiltrated inflammatory cells; blue arrow: hepatocyte disintegration). Scale bar: 50 μm. * *p* < 0.05 and ** *p* < 0.01 vs. the CG. ## *p* < 0.01 vs. the AG.

**Figure 6 foods-12-01006-f006:**
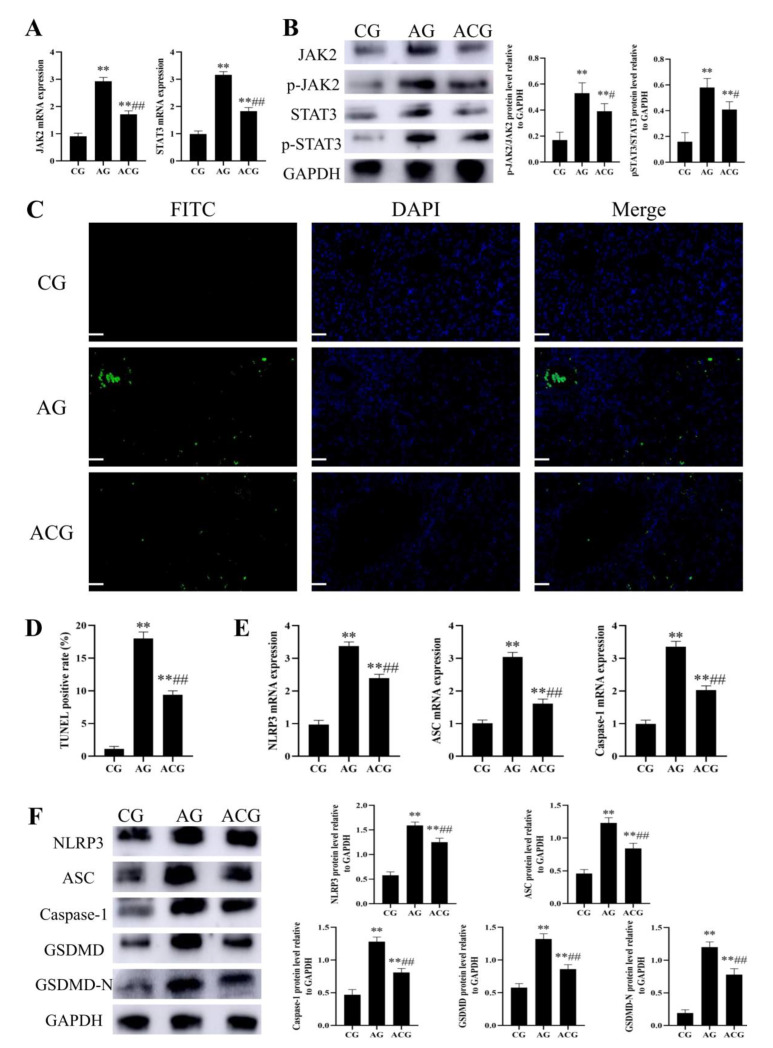
Curcumin inhibited JAK2/NLRP3-mediated pyroptosis of duck livers caused by AFB_1_. (**A**) mRNA expressions of JAK2 and STAT3 (*n* = 6). (**B**) Western blot of the JAK2, p-JAK2, STAT3, and p-STAT3 protein (*n* = 3). (**C**) Representative images of liver TUNEL staining. Scale bar: 50 μm. (**D**) TUNEL analysis of livers (*n* = 3). (**E**) mRNA expressions of IL-1β and IL-18 (*n* = 6). (**F**) Western blot analysis of the NLRP3, ASC, Caspase-1, GSDMD, and GSDMD-N protein levels (*n* = 3). ** *p* < 0.01 vs. the CG. # *p* < 0.05 and ## *p* < 0.01 vs. the AG.

**Figure 7 foods-12-01006-f007:**
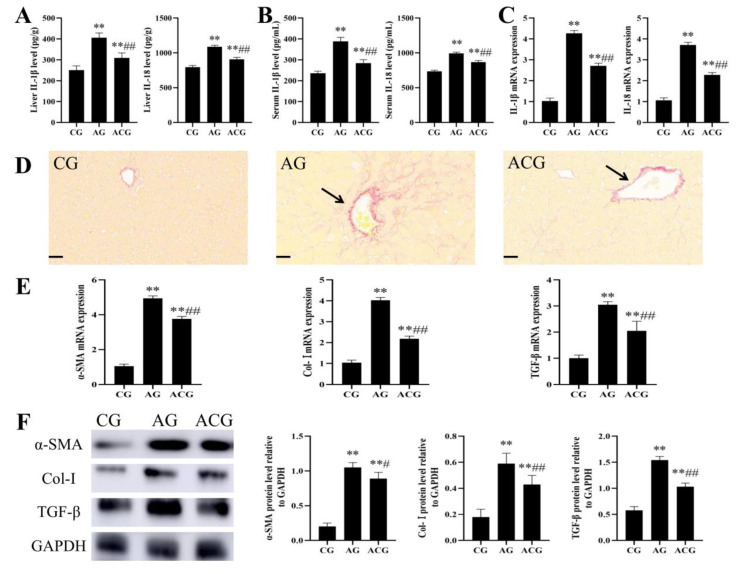
Curcumin alleviated liver fibrosis in ducks caused by AFB_1_. (**A**) Liver IL-1β and IL-18 levels (*n* = 6). (**B**) Serum IL-1β and IL-18 levels (*n* = 6). (**C**) mRNA expressions of IL-1β and IL-18 (*n* = 6). (**D**) Representative images of sirius red staining of duck livers (black arrow: fibrotic lesions). (**E**) mRNA expressions of α-SMA, Col-I, and TGF-β (*n* = 6). (**F**) Western blot analysis of the α-SMA, Col-I;, and TGF-β protein levels (*n* = 3). ** *p* < 0.01 vs. the CG. # *p* < 0.05 and ## *p* < 0.01 vs. the AG.

## Data Availability

The data used and analyzed in this study are available from the corresponding author on reasonable request.

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
