# Peer review of "Curcumin Alleviates Aflatoxin B1-Induced Liver Pyroptosis and Fibrosis by Regulating the JAK2/NLRP3 Signaling Pathway in Ducks"

_foods, 2023, doi:10.3390/foods12051006_

Round 1
Reviewer 1 Report
It was very interesting to read your article and your conclusions are succinct. However, the purpose of the study is not entirely clear regarding the dimension of economic or social loss that aflatoxin might cause in the poultry sector, ducks in your case, and or in humans.
It is not clear whether the experiment took place as part of a research institute and or university; the location of the experiment is also missing. It is also unclear the conditions the animals were raised.
It seems that Curcumin was added to the feed as a complement and it was not an ingredient of the feed per se. Perhaps in future studies you could consider the viability of adding Curcumin as a feed ingredient and any likely issues of loss of functionality as a result of feed processing.
Author Response
Response to Reviewer 1 Comments
Point 1: It was very interesting to read your article and your conclusions are succinct. However, the purpose of the study is not entirely clear regarding the dimension of economic or social loss that aflatoxin might cause in the poultry sector, ducks in your case, and or in humans.
Response 1: Thanks for the reviewer's suggestion. We have added relevant literature on the economic losses caused by aflatoxin B1 to the poultry and society in the Introduction.
Point 2: It is not clear whether the experiment took place as part of a research institute and or university; the location of the experiment is also missing. It is also unclear the conditions the animals were raised.
Response 2: Sorry for what we lacked in the manuscript. Our experiments were done in the university, the location of the experiment and the conditions in which the animals were kept have been added to the Materials and methods.
Point 3: It seems that Curcumin was added to the feed as a complement and it was not an ingredient of the feed per se. Perhaps in future studies you could consider the viability of adding Curcumin as a feed ingredient and any likely issues of loss of functionality as a result of feed processing.
Response 3: Thanks for the reviewer's suggestion. Indeed, curcumin in this study was added to feed as a supplement, and we found that some scholars used curcumin as a component of feed for production [1]. We will conduct more exploration in this aspect in the future.
- Campigotto, G.; Alba, D.F.; Sulzbach, M.M.; Dos Santos, D.S.; Souza, C.F.; Baldissera, M.D.; Gundel, S.; Ourique, A.F.; Zimmer, F.; Petrolli, T.G.; et al. Dog food production using curcumin as antioxidant: effects of intake on animal growth, health and feed conservation. Archives of animal nutrition 2020, 74, 397-413, doi:10.1080/1745039x.2020.1769442.
Reviewer 2 Report
The manuscript entitled "Curcumin alleviates aflatoxin B1-induced liver pyroptosis and fibrosis by regulating the JAK2/NLRP3 signaling pathway in ducks" is an interesting work; it is a meaningful contribution to the field of mycotoxins in animal. In addition, the paper has scientific rigor, and it is well writing, reads well and is concise. Original work with a great interpretation of the results without overreaching. Although, I suggest some little changes for its improvement.
Abstract. Please define the abbreviations for the janus kinase 2, signal transducers and activators of transcription 3 and nod-like receptor protein 3 (JAK2, STAT3 and NLRP3) immediately before their first utilization.
Introduction. If you consider that "Aspergillus parasitica" is synonymous with "Aspergillus parasiticus", please use this second name because it is much most common.
Discussion. If you consider that the values obtained for the analytes assessed during the first and second part of the experiment did not present significant differences, please address the comparison of the equivalent groups (CG HG vs CG AG) to highlight both the stability of the test methodologies and the merit of curcumin in the ACG group.
Author Response
Response to Reviewer 2 Comments
Point 1: Abstract. Please define the abbreviations for the janus kinase 2, signal transducers and activators of transcription 3 and nod-like receptor protein 3 (JAK2, STAT3 and NLRP3) immediately before their first utilization.
Response 1: Thanks for the reviewer's suggestion. We have revised these questions in the Abstract.
Point 2: Introduction. If you consider that "Aspergillus parasitica" is synonymous with "Aspergillus parasiticus", please use this second name because it is much most common.
Response 2: Thanks for the reviewer's suggestion. As the editor pointed out that the manuscript contained too much background information on aflatoxin, we have deleted this sentence in the revised version. In the future, we will pay attention to the correct spelling of "Aspergillus parasiticus".
Point 3: Discussion. If you consider that the values obtained for the analytes assessed during the first and second part of the experiment did not present significant differences, please address the comparison of the equivalent groups (CG HG vs CG AG) to highlight both the stability of the test methodologies and the merit of curcumin in the ACG group.
Response 3: Thanks for the reviewer's suggestion, we have added this content in the Discussion.
Reviewer 3 Report
The authors investigated whether curcumin could alleviate AFB1-induced liver pyroptosis and fibrosis by regulating the JAK2/NLRP3 signaling pathway in ducks, so as to provide experimental basis for alleviating liver toxicity of AFB1 in clinic. The authors employed very useful methods, and shown promising findings. However, there are areas of this work that warrants further deliberations
a) As this work is about curcumin, introduction especially a paragraph before the last should have: i) curcumin, its chemical constituents and where it is naturally abundant; ii) What are the efficacious potentials of curcumin from scientific literature; iii) Any published report about cucrmin, especially its metabolic impact, what is lacking in relevant literature.
b) Please start the matierals and methods with a new sub-section captioned "Schematic Overview of Experimental Progam", which should comprise at least 4 sentieces, supported by a flow diagram. Make sure you introduce this section with the figure, explaining the various steps, from identification/selection of samples, up to how there were allocated to various tested parameters.
d) The reviewer has read the results and discussion. It is promising for now. Please, kindly insert in the discussion "(Refer to Figure x)" in all the places where specifically those were being discussed. That is to say, by the time one reads the discussion, one must have captured all the figures displayed in the results.
Look forward to your revised manuscript.
Author Response
Response to Reviewer 3 Comments
Point 1: As this work is about curcumin, introduction especially a paragraph before the last should have: i) curcumin, its chemical constituents and where it is naturally abundant; ii) What are the efficacious potentials of curcumin from scientific literature; iii) Any published report about cucrmin, especially its metabolic impact, what is lacking in relevant literature.
Response 1: Thanks for the reviewer's suggestion, we have added this content in the Introduction.
Point 2: Please start the matierals and methods with a new sub-section captioned "Schematic Overview of Experimental Progam", which should comprise at least 4 sentieces, supported by a flow diagram. Make sure you introduce this section with the figure, explaining the various steps, from identification/selection of samples, up to how there were allocated to various tested parameters.
Response 2: Thanks for the reviewer's suggestion, we have added the "Schematic Overview of Experimental Progam"(Figure 1) in the Matierals and methods.
Point 3: The reviewer has read the results and discussion. It is promising for now. Please, kindly insert in the discussion "(Refer to Figure x)" in all the places where specifically those were being discussed. That is to say, by the time one reads the discussion, one must have captured all the figures displayed in the results.
Response 3: Thanks for the reviewer's suggestion, we have added this content in the Discussion.
Round 2
Reviewer 3 Report
The authors have made substantial efforts and improved the work.
The reviewer believes the work is acceptable for publication.
Author Response
感谢您帮助我们改进题为“姜黄素通过调节鸭的JAK1 / NLRP2信号通路来缓解黄曲霉毒素B3诱导的肝脏焦亡和纤维化”的手稿。我们已根据编辑的建议修改了稿件。